# Influence of the Text Neck Posture on the Static Dental Occlusion

**DOI:** 10.3390/medicina58091303

**Published:** 2022-09-18

**Authors:** Eniko Tunde Stoica, Corina Marcauteanu, Anca Tudor, Virgil-Florin Duma, Elena Constanta Amaricai, Roxana Onofrei, Oana Suciu, Meda Lavinia Negrutiu, Cosmin Sinescu

**Affiliations:** 1TADERP Research Center, “Victor Babes” University of Medicine and Pharmacy of Timisoara, 9 Revolutiei 1989 Ave., 300070 Timisoara, Romania; enikodemjan@yahoo.com; 2School of Dental Medicine, “Victor Babes” University of Medicine and Pharmacy of Timisoara, 2A Eftimie Murgu Place, 300070 Timisoara, Romania; atudor@umft.ro (A.T.); meda_negrutiu@yahoo.com (M.L.N.); minosinescu@gmail.com (C.S.); 3Research Center in Dental Medicine Using Conventional and Alternative Technologies, School of Dental Medicine, “Victor Babes” University of Medicine and Pharmacy of Timisoara, 9 Revolutiei 1989 Ave., 300070 Timisoara, Romania; amaricai.elena@umft.ro; 43OM Optomechatronics Group, Faculty of Engineering, “Aurel Vlaicu” University of Arad, 2 Elena Dragoi Str., 310177 Arad, Romania; 5Doctoral School, Polytechnic University of Timisoara, 1 Mihai Viteazu Ave., 300222 Timisoara, Romania; 6Department of Rehabilitation, Physical Medicine and Rheumatology, Research Center for Assessment of Human Motion, Functionality and Disability, “Victor Babes” University of Medicine and Pharmacy of Timisoara, 2A Eftimie Murgu Place, 300070 Timisoara, Romania; onofrei.roxana@umft.ro (R.O.); oanasuciu78@umft.ro (O.S.)

**Keywords:** text neck posture (TNP), static occlusion, maximum intercuspation (MI), T-Scan III system, occlusion time (OT), asymmetry index of the occlusal force (AOF)

## Abstract

*Background and Objectives:* The excessive use of smartphones for various tasks led to a new adverse postural phenomenon called text neck. The aim of this study was to investigate the effect of the text neck posture (TNP) on static occlusion by using the T-Scan III occlusal diagnostic system. *Materials and Methods*: Nineteen subjects (aged 20 to 24 years) were considered for this research. They had normal values for anterior overbite and overjet, Angle Class I occlusion, no posterior crossbite, and no signs or symptoms of cervical or temporo-mandibular disorders. Occlusal registrations were performed with the T-Scan III system in a normal, neutral head posture (NHP), as well as in the TNP. The investigated parameters were: occlusion time (OT), asymmetry index of the occlusal force (AOF), percent of the maximum movie force (%MMF), and the time elapsed from the last occlusal contact until the maximum intercuspation (MAT-OTB). The last three parameters were analyzed in the maximum area frame (MA) of the registrations. For the statistical analysis of the recorded data, the Wilcoxon Signed Ranks test and the Spearman’s correlation coefficient were used. *Results:* The following values were obtained in NHP and in TNP: for AOF, 14.88 ± 10.39% and 18.04 ± 12.83%, respectively; for OT, 1.34 ± 1.84 s and 1.32 ± 1.8 s, respectively; for the %MMF, 97.5 ± 2.83% and 96.31 ± 3.17%, respectively; for MAT-OTB, 2.08 ± 1.82 s and 1.45 ± 2.3 s, respectively. There were no statistically significant differences between the static occlusal parameters measured in NHP and those in TNP. However, the high values of the AOF and OT in NHP revealed an imbalance of the occlusal force distribution between the right and left side in maximum intercuspation (MI), as well as a lack of simultaneity of static occlusal contacts. Furthermore, there was a significant, direct, and strong correlation between OT and AOF in NHP. *Conclusions:* The NHP should not be used as the starting position in TNP simulations in T-Scan studies, so as to avoid statistically insignificant differences between static occlusion in NHP and TNP. The healthy standing subjects, with normal occlusal relationships from the clinical point of view, revealed an occlusal instability in NHP when examined with the T-Scan.

## 1. Introduction

The term *text neck* was proposed by a chiropractor, Dr. Dean L. Fishman [1], and it is used to define both a bad postural position and a syndrome associated with the prolonged and inappropriate use of handheld mobile devices, including smartphones [2,3,4]. This adverse postural phenomenon can be described as a sustained flexed neck position, with the head tilted forward. It is associated with forward-rolled shoulders, which increase the curve of the thoracic spine [2,4]. Compared to the neutral posture, the higher neck flexion angle requires an increased activity of the neck muscles in order to compensate for the effect of gravity [5]. These biomechanical changes in the cervical and thoracic spine, as well as muscular imbalances and postural compensations, finally lead to cervical muscle fatigue and pain [2,3].

The scientific literature demonstrates that any change in the head and neck posture induces a change in the rest position of the mandible [6], in the activity of the masticatory muscles [7], and in the habitual path of mouth closing [8]. In an interesting study, Yamada et al. (1999) found that as the head bended forward (i.e., in ventroflexion), the closing path approached the *maximum intercuspation position* (MIP) from the anterior region [9]. The forward bending of the head also decreased the stability of the closing path. On the other hand, as the head was bent backward, the closing path approached the MIP from the posterior region and its stability increased.

A widely debated topic in the literature is whether or not changes in head and neck posture have a significant influence on dental occlusion. However, the effect of *text neck posture* (TNP) on occlusion has been overlooked in the dental literature. Chapman et al. (1991) used the T-Scan system to record and analyze the occlusal contacts that occur in *maximum intercuspation* (MI) in three different head positions: supine, sitting erect, and sitting with the head tipped forward [10]. They proved that the mandible is pushed forward during closure when the head is tipped forward, resulting in more anterior initial contacts. However, the total number of occlusal contacts in MI was not significantly modified by the postural change. This means that the main effect of a changed head position was on the initial tooth contacts, which guide the mandible back into MI from eccentric positions. The authors speculated that the location of the first occlusal contacts in the anterior region of the dental arches during closure (i.e., when the head is tipped forward) could produce occlusal overload of the individual teeth. Therefore, they are expected to alter the activity of the elevator muscles.

In another T-Scan study of mouth closure, Makofsky et al. (1991) found that in subjects 30 years of age and older, a 30° ventroflexion of the head shifted the initial occlusal contacts anteriorly, while a 45° head extension displaced the contacts posteriorly older [11]. Gupta et al. (2017) reported that the occlusal contact area in MI varies between two different head postures: 90° upright and 30° ventroflexed [12]. The pressed occlusal contact area (mm^2^) was measured using the Dental Prescale System (Dental Prescale, Fuji Film Co., Tokyo, Japan), a computerized occlusal analysis system used for the measurement and analysis of the bite force (N), the occlusal contact area (mm^2^), and the bite pressure (MPa). They concluded that the pressed occlusal contact area (mm^2^) decreased in head ventroflexion compared to the upright-erect position.

Such studies have proved that physiological head positions, which are adopted naturally by the subjects, have a significant influence on some of the investigated static occlusal parameters, including the number and position of the initial tooth contacts during mouth closure, as well as on the occlusal contact area in MI. It must be pointed out that a 30° ventroflexion of the head coincides with the active feeding posture, while a 45° head extension is used for drinking [13].

Fewer studies have addressed the effect of abnormal head positions such as the *forward head posture* (FHP) on dental occlusion. In FHP, the subject pushes her/his head in front of its natural position over the cervical spine, with a simultaneous posterior bending of the head and a compensatory extension of the upper cervical spine, in order to maintain the horizontal direction of the eyes—for example, facing the computer desktop [14] (Figure 1a). Some authors even call this the desktop neck posture [15].

In a sample of thirty-nine normal subjects, Makofsky (2000) did not find a significant relationship between experimentally induced FHP and the initial occlusal contact pattern that occured while the subject was slowly and completely biting onto the sensor of the T-Scan II Occlusal Diagnostic System [16]. This result is of interest because the same author proved in a previous study that the initial occlusal contact pattern during closure is influenced by the extension of the head [11]. It appears that in FHP, which involves a significant degree of head and upper cervical spine extension, the alteration in the mandibular position is not important enough to produce the occlusal changes observed during a physiological head extension. This conclusion has clinical relevance in the treatment of patients with *temporo-mandibular disorders* (TMD) and FHP. Some authors claim a relationship between FHP and TMD [17,18]. If the initial occlusal contact pattern does not change in FHP, that means that the occlusion is not an etiological factor of the TMD and should not be therapeutically addressed.

It must be pointed out that FHP is different from the TNP (as pointed out in Figure 1), although they are often incorrectly used as synonyms. Assumed while using a smartphone, the TNP is characterized by a flexed position of the upper cervical region, with the eyes facing downward and fixed on the smartphone [2]—Figure 1b.

Following the impact of this aspect on today’s population, as well as the shortcomings of the literature on this topic, as pointed out above, the aim of this study was to investigate the influence of TNP on static occlusion in young healthy subjects with normal occlusal relationships.

## 2. Materials and Methods

The present study was conducted according to the guidelines of the Declaration of Helsinki. It was approved by CECS no. 70/22.12.2021 of the Ethical Committee of the “Victor Babes” University of Medicine and Pharmacy of Timisoara, Romania.

### 2.1. Study Subjects

A total of nineteen subjects, fifteen females and four males (aged 20 to 24 years), were considered for this research. This sample was chosen for convenience. The process of consecutive selection among young subjects who were interested in a free examination of their masticatory system was based on the following inclusion criteria: full dental arches (except for the third molars in some subjects); normal values of overbite (2 to 4 mm) and overjet (1 to 2 mm); Angle Class I occlusion, without posterior crossbite; healthy periodontal status, with normal physiological tooth mobility; no signs and symptoms of TMD; no pain or limited range of motion in the cervical spine.

In addition, all standing subjects were able to adopt a *normal head posture* (NHP), with the head and back straight and with the external auditory meatus on the same vertical line as the acromio-clavicular joint (i.e., the shoulder), the hip, and the knee. Their head was placed in the midline, with the chin above the manubrium, while their neck had a slight lordotic curve and a normal length, without tilting or rotation of the head [19].

The exclusion criteria were: degenerative or inflammatory spine pathology; fibro-myalgia; cervical spine traumatic events; autoimmune diseases; and neurologic diseases that can have an impact on cervical position. Furthermore, we excluded subjects that currently reported any of the following parafunctions, which can influence the head and neck position by muscle hyperactivity: awake and/or sleep bruxism; nail biting; biting of the cheeks and lips; ventral position during sleep; keeping the phone between the face and the shoulder; playing violin or a wind instrument; maintaining a pencil between the dental arches; tongue pressing on the lingual surfaces of teeth; abusive consumption of chewing gum [20].

The anamnesis and clinical examination of the masticatory system was based on Schiffman’s “Diagnostic Criteria for Temporomandibular Disorders (DC/TMD) for Clinical and Research Applications” [21]. The examination of the cervical area was based on the protocol recommended by Ombregt [19].

All subjects were fully informed about the nature of the investigation and signed an informed consent form to participate in this research.

### 2.2. Examination Procedure

At the beginning of the procedure, we explained the different steps of the examination to our subjects without giving any details about the purpose or hypotheses of the research. The images in Figure 2 and Figure 3 present one of the examiners as she demonstrates the NHP (Figure 2) and the TNP (Figure 3) to the study subjects.

Further on, the subjects were instructed on how to adopt an NHP. Then they were asked to text for 60 s on their mobile phones, in order to relax and get accustomed to the TNP. For most of them, the mandible went into the postural position (with the freeway space between the dental arches) during texting, but each one of them swallowed at some point in MI, which was the position we followed in our research. The data from the dental literature showed a mean spontaneous swallowing frequency of 0.98 swallows/minute for healthy young subjects [22].

Each occlusion was objectively assessed by the same examiner by using the T-Scan III system (Tekscan Inc., Ann Arbor, MI, USA), following a standardized protocol. Thus, the standing subject was asked to hold the scanning handle and to adopt an NHP, with the head and back straight and with the external auditory meatus on the same vertical line as the acromio-clavicular joint (i.e., the shoulder), the hip, and the knee. A second examiner measured *the neck flexion angle* of the subject in NHP with a manual goniometer. This neck flexion angle is defined as the angle between a vertical line raised from the C7 spinous process and a line connecting the C7 spinous process to the mid-tragus [5]. The subject was instructed to forcefully clench on the sensor while the T-Scan occlusal registration was made. No pain was elicited in the masticatory system or in the cervical area. In the following phase, each subject was asked to further flex his neck forward, till she/he reached a neck flexion angle that was 30° higher than the angle measured in NHP. A new T-Scan registration was made in this TNP.

The digital occlusal analysis of each patient was preceded by mock registrations that allowed the patient to get accustomed to the procedure. At the same time, we could adjust the sensitivity of the T-Scan III system in order to ensure that its force recording range was matched to the “bite strength” of each individual patient [23]. We changed the sensor before taking the actual recordings.

The T-Scan III movies were saved and labeled in a way that blinded the data collectors who analyzed those movies. Thus, they did not know in which head and neck position each of the recordings was made.

### 2.3. Data Collection and Statistical Analysis

The distribution, force, and timing of occlusal contacts were analyzed in MI and in static intercuspation. The latter was defined by Kerstein as the moment of the last tooth contact during the closure of the mouth, which is marked as the Bline on the registration. It is different from the MI, which appears later [23,24,25]. The maximum intercuspation was analyzed in the maximum area frame of the registration (MA). In an ideal occlusion, the MA frame is also the frame where the *maximum movie force* (MMF) occurs.

The first analyzed occlusal parameter was *occlusion time* (OT)—Figure 4b and Figure 5b—defined by Kerstein as the elapsed time (in seconds) measured from the first occlusal contact (A line) until the last tooth contact in static intercuspation (B line) during mouth closure [24,25]. The T-Scan III system was set at a normal OT value of under 0.3 s (“within range”); the “borderline range” was 0.3 to 0.5 s [23].

In the MA frame (MI), the following aspects were investigated (Figure 4 and Figure 5):(i)*Distribution of the occlusal force* between the left and right SSS side of the arch (Figure 4a and Figure 5a). The ideal occlusion has 50% of the occlusal force on each side, but the T-Scan III System Manual states that a distribution of 53% to 47% of the occlusal force can be considered within the normal range [23]. Based on these values, we *calculated the asymmetry index of the occlusal force* (AOF) according to the formula [26]:**AOF (%) = [(occlusal force on the right side−occlusal force on the left side)/total occlusal force] × 100**
AOF has normal values ranging from 0 to 6%.(ii)*Percent of the maximum movie force* (%MMF)—Figure 4b and Figure 5b. In an ideal occlusion, the MA frame is the frame with the largest area of tooth contact, but also the frame with the MMF.(iii)*Time elapsed from the last occlusal contact in static intercuspation till MI* (MAT-OTB). It describes the time that passes till the slopes of the guiding cusps bring the mandible into MI. This parameter was calculated by subtracting the OTB value from the MAT value. Both values are displayed by the T-Scan software (Figure 4b and Figure 5b).

Statistical processing was performed using the SPSS 17.0 software package. Descriptive statistics were calculated for several continuous variables: OT, AOF, %MMF, and MAT-OTB. The comparisons between the two paired numerical series were performed using the non-parametric Wilcoxon Signed Ranks Test. The results were considered significant for a value of *p* < 0.05. The correlations between variable pairs were made by calculating the Spearman’s correlation coefficient, since they did not have a normal distribution. The correlations were considered significant for a value of *p* < 0.05, as well.

## 3. Results

The demographic characteristics of the study group are presented in Table 1.

The values of the OT were 1.34 ± 1.84 s in NHP and 1.32 ± 1.8 s in TNP, while the difference between the two head and neck positions was small and not statistically significant (*p* = 0.658), as presented in Table 2. The values of the AOF were 14.88 ± 10.39% in NHP and 18.04 ± 12.83% in TNP; the increase in AOF in TNP was not statistically significant (*p* = 0.344)—Table 2. The values of the MAT-OTB were 2.08 ± 1.82 s in NHP and 1.45 ± 2.3 s in TNP. Therefore, the decrease in MAT-OTB in TNP was not statistically significant (*p* = 0.117)—Table 2. The values of the %MMF were 97.5 ± 2.83% in NHP and 96.31 ± 3.17% in TNP; the difference between the two head and neck positions was small and not statistically significant (*p* = 0.251). The fact that the studied parameters’ variations between the two head and neck positions were not statistically significant could be due to the choice of the NHP as the starting position of our T-Scan study. Adding 30° to the neck flexion angle measured in NHP could have brought the handle and the power cable of the T-Scan into an inappropriate position, leading to the collection of TNP parameter values that were not conclusive in some subjects.

Although NHP was inappropriate as a starting point to simulate the TNP in our T-Scan study, the recordings made in that position revealed some significant and useful results. Thus, the mean value of the AOF was 14.88 ± 10.39% in NHP. Perfect symmetry of the occlusal force in MI was not found in any subject, nor in the NHP. Instead, we found an AOF less than or equal to 6% in four subjects with NHP (i.e., subject number 15, 16, 17, and 19; 21.05%). All other subjects had AOF values above 6%, which indicate an unbalanced distribution of the occlusal forces between the right and left side of the dental arches in NHP. The actual distribution of occlusal forces between the left and right side of the arch in each subject can be observed in Appendix A.

The mean value of the OT was 1.34 ± 1.84 s in NHP. Bilateral simultaneous occlusal contacts in static intercuspation are generally associated with an OT under 0.2 s [24,25] or 0.3 s [23]. In NHP, we found an OT below 0.3 s in only four subjects (i.e., subject number 7, 8, 17, and 19; 21.05%). All other subjects had OT values above 0.3 s, which revealed a lack of simultaneity of the occlusal contacts in static intercuspation. The OT values of all subjects can be viewed in Appendix A.

The correlation between OT and AOF was significant, direct, and strong in NHP (Spearman’s rho = 0.724, *p* < 0.001)—Figure 6. An increased OT was associated with an increased AOF. The lack of simultaneity of the occlusal contacts in static intercuspation (as concluded from high OT values), associated with an unbalanced distribution of the occlusal forces between the right and left side of the dental arches in MI (high AOF values), point to an unstable static occlusion in NHP.

In an ideal occlusion, the %MMF used by subjects to maintain MI in the MA frame is 100%, but we found such a value only in subject number 16 while she maintained an NHP. The value of the %MMF was 97.5 ± 2.83% in NHP. It is high enough to be considered acceptable, given the fact that ideal occlusion is more of a theoretical concept than a clinical reality. The correlations between %MMF and the other studied parameters were not conclusive. OT was insignificantly, inversely, and weakly associated with %MMF (Spearman’s rho = −0.168, *p* = 0.492), while AOF and %MMF had a similar behavior (Spearman’s rho = −0.164, *p* = 0.502).

A new parameter followed in this study was MAT-OTB, which refers to the time elapsed from the last occlusal contact in static intercuspation (B line) to MI, characterized by the maximum area of tooth contact (MA frame). The value of the MAT-OTB was 2.08 ± 1.82 s, measured in NHP in standing subjects (Table 2). Further studies are necessary in order to establish the normal values of this parameter, in conditions similar to those utilized by Kerstein to determine the normal values of OT and AOF.

Notations: OTN, AOFN, MMFN, and MAT-OTBN are the values of the variables in NHP; OT30, AOF30, MMF30, and MAT-OTB30 are the values of the variables in TNP.

The correlations between MAT-OTB and the other three studied parameters were not conclusive. The correlation between OT and MAT-OTB was insignificant, inverse, and weak (Spearman’s rho = −0.154, *p* = 0.530). The same was valid for the correlation between MAT-OTB and %MMF (Spearman’s rho = −0.165, *p* = 0.499). The correlation between MAT-OTB and AOF was non-linear and insignificant (Spearman’s rho = 0.043, *p* = 0.861).

## 4. Discussion

In the present study, we investigated the influence of the TNP on static occlusion. TNP is associated with the prolonged and inappropriate use of handheld mobile devices, including smartphones [2,3,4]. It is a topic of interest due to the large-scale use of smartphones for various tasks, including texting. For example, university students aged 19 to 22 years from the USA reported using a handheld mobile device for over 8.5 h/day on average [27]. It appears that the extended duration and frequency of smartphone use, called the “invisible addiction” by Roberts et al. (2014), is correlated in many young people with the fear of missing out (FoMO) [27,28]. FoMO can be described as the perception that others might be having a pleasing experience from which one is absent, followed by a compulsive behavior to maintain these social connections [28,29].

In the dental literature, the correlation between TNP and occlusion was less studied compared to the effect of the FHP on occlusal contacts. Both TNP and FHP represent bad postures which can cause neck pain and disorders of the craniocervical complex, but they are different from each other [2,3]. Using them as synonyms can lead to conflicting and confusing results in the literature.

In the present study, we paid special attention to the placement of the subjects in TNP, which is characterized by the flexion of the upper spine, with the eyes looking downward at an “imaginary smartphone” [2]. One of our examiners repeatedly demonstrated the TNP to the subjects involved in the study. We tried to avoid the posterior bending of the head and the compensatory extension of the upper cervical spine in order to maintain a horizontal eye line [14], which is characteristic during FHP. The investigation was further complicated by the fact that the subjects were standing during the T-Scan recordings in both NHP and TNP.

In order to standardize the TNP, we used the neck flexion angle, measured with a manual goniometer. The body of the goniometer was centered on the C7 spinous process, with the vertical fixed arm pointing upwards and the middle of the mobile arm passing through the mid-tragus [2]. The first measurement was done in NHP, with the subjects standing and their back straight. We added another 30° to the initial value of the neck flexion angle in order to simulate TNP.

It must be pointed out that the literature does not offer a clear value of the neck flexion angle that represents the “tipping point” from a NHP to a TNP. This depends very much on the devices used for the angle measurement. Ailneni et al. (2019) considered poor neck posture to start at a neck flexion angle larger than 15° relative to the neutral posture [5]. They chose a 15° flexion as a threshold since Hansraj (2014) found in a previous study that it induced a two to three times greater stress in the neck compared to the neutral posture [30]. In the study performed byAilneni et al. (2019), kinematics data of the head and neck were collected by an optical motion capture system with six Flex 13 infrared cameras (Natural Point, Corvallis, OR, USA) [5]. Namwongsa et al. (2019) proved that a neck flexion angle increased over 15° during texting on a smartphone is associated with higher activity in the cervical erector spinae muscles, which can lead to neck pain. A Cervical Range of Motion (CROM) device (Performance Attainment Associates, St. Paul, MN, USA) was used to measure the neck flexion angle [31]. The devices used in the above studies to measure the neck flexion angle were more accurate than our manual goniometer, and this is a limitation of our study.

In the present study, we considered the TNP to start at a neck flexion angle larger than 30° relative to the NHP. It is interesting to evaluate the changes that occur at a degree of flexion that can elicit neck pain when abused. In this respect, Kim et al. (2015) stated that while performing a texting task on the smartphone, individuals with minor neck pain tend to bend their neck slightly more than individuals without neck pain [32].

Lee et al. (2016) demonstrated that body posture and the duration of the smartphone usage influence the neck flexion angle in healthy young subjects [33]. They found that the neck flexion angle was higher in the standing position than in the sitting on a chair and sitting on the floor positions. This is the reason why we made our T-Scan recordings on standing subjects in both NHP and TNP.

In the present study, we investigated the influence of the TNP on static intercuspation (as defined by Kerstein and Grundset 2001 [24]) and MI for several reasons: the use of smartphones for different tasks (e.g., texting, internet surfing, etc.) increases the odds of awake bruxism, which consists in the forceful, involuntary, and unconscious clenching of teeth in MI [34,35]; the occlusal instability in MI can lead to an overload of the masticatory system and consequently to TMD if the adaptation capacity of the masticatory system is exceeded [26,36,37]; deglutition occurs in MI in most subjects [38].

We examined the static occlusal parameters by using the T-Scan III system, which proved to be accurate and reliable for the digital analysis of occlusal contact distribution [39], as well as timing and force [40], starting from the initial tooth contact to MI. This device ensures a more detailed and precise occlusal analysis than traditional articulating paper, which can only determine the location of the occlusal contacts. In contrast, the T-Scan III provides information about the distribution, timing, and force of those contacts during the entire closure movement [41,42].

The AOF and MAT-OTB parameters that were examined in this study in NHP and in TNP describe if the subject has a stable MI or not. The OT is evaluated in static intercuspation, which practically represents an intermediate station in the path of the mandible towards MI. Furthermore, it is important in achieving occlusal stability.

*The AOF* illustrates the distribution of occlusal force between the left and right side of the arch and is an indicator of the balance of occlusal contacts in MI [23,24,25]. As the values of the AOF were 14.88 ± 10.39% in NHP and 18.04 ± 12.83% in TNP, one may remark that the AOF increase in TNP was not statistically significant (*p* = 0.344). As stated in the results section, this could be due to the choice of the NHP as the starting position of our T-Scan study. Adding 30° to the neck flexion angle measured in NHP could have brought the handle and the power cable of the T-Scan into an inappropriate position. That could have led to the collection of TNP parameter values that were not conclusive in some subjects. This is why we do not recommend the use of NHP as a starting point to simulate the TNP in T-Scan studies. To enhance the design of further T-Scan studies, we propose using a head posture with a neck flexion angle of 0° as a starting point to simulate the TNP and a Cervical Range of Motion device to measure that angle.

The mean value of the AOF was 14.88 ± 10.39% in NHP. It is below the AOF limit that is considered by some studies in the dental literature to be an occlusal risk factor associated with TMD. For Wang et al. (2012), the “tipping point” towards the myogenous or arthrogenous TMD was an AOF of 16.66 ± 0.47% [26]. Dzingutė et al. (2017) found a significantly higher AOF in twenty patients with pain in the temporo-mandibular joints (15.90 ± 2.71%) than in the twenty-four subjects of the healthy control group (12.93 ± 1.88) [36]. In conclusion, our subjects do not have an increased risk for TMD based on the medium value of the AOF measured in NHP.

The AOF was less than or equal to 6% in only four subjects with NHP, which represents the normal value assigned to this parameter in the T-Scan III System Manual, corresponding to a balanced distribution of the occlusal force in MI. All other subjects had AOF values above 6%. This means that most of our subjects had an imbalance of the occlusal force distribution between the right and left side in MI in NHP. That imbalance existed despite their normal occlusal relationships (shown by the normal values of the anterior overbite and overjet, the Angle Class I occlusion, and the lack of posterior crossbite) and was still compensated by the masticatory system, because none of our subjects were accused of signs and/or symptoms of TMD.

*The OT* was defined by Kerstein et al. (2001, 2015) as the elapsed time in seconds measured from the first occlusal contact until the last tooth contact in static intercuspation during mouth closure [24,25]. The T-Scan III system was set at a normal OT value of under 0.3 s (“within range”); the “borderline range” was 0.3 to 0.5 s [23], as pointed out in Section 2. The OT describes the degree of bilateral simultaneity of the static occlusal contacts. Having bilateral simultaneity is a desirable state of occlusal health, as well as a requirement for an optimum occlusal design and for a normal functional occlusion. In the present study, the OT values were 1.34 ± 1.84 s in NHP and 1.32 ± 1.8 s in TNP. The difference between them was not statistically significant for the same reasons that led to statistically insignificant AOF values in TNP.

In the present study, we obtained an OT mean value of 1.34 ± 1.84 s in NHP. That value was higher than the one recommended by Kerstein and Grundset (2001) or found in other T-Scan studies on subjects with normal occlusal relationships and no TMD [24,26]. In one such study, Wang et al. (2012) reported an OT of 0.69 ± 0.03 s for healthy subjects [26], while in a similar research study, Baldini (2015) found a value of 0.45 ± 0.17 s [37]. On the other hand, the OT in NHP had lower values than the OT found by Wang et al. (2012) for TMD patients with normal occlusal relationships (2.05 ± 0.06 s) [26], but higher values than the OT reported by Baldini (2015) for dysfunctional patients (0.64 ± 0.21 s) [37]. Baldini assumed that he used patients with less severe TMDs in his study, as compared to those analyzed by Wang et al. (2012). In conclusion, our subjects do have an increased risk for a less severe TMD based on the medium value of the OT measured in NHP in our study, as well as the values advanced by Baldini for dysfunctional patients [37].

In NHP, we found an OT below 0.3 s in only four subjects. All other subjects had OT values above 0.3 s, which revealed a lack of simultaneity of the occlusal contacts in static intercuspation, despite their normal occlusal relationships from a clinical point of view. That lack of simultaneity is still compensated by the masticatory system of our healthy subjects.

The high values of the AOF and OT in NHP describe an occlusal instability that is further amplified by the fact that the correlation between OT and AOF is significant, direct, and strong in NHP. An imbalance of the occlusal force distribution between the right and left side in MI, which coexists with a lack of simultaneity of occlusal contacts in static intercuspation, could put the adaptability of the masticatory system to a hard test. Because our patients do not have signs or symptoms of TMD disorders, occlusal adjustment by selective grinding is not indicated [20].

*Values of the %MMF*, obtained as 97.5 ± 2.83% in NHP and 96.31 ± 3.17% in TNP, showed that the difference between the two head and neck positions was small and not statistically significant. In addition, the correlation between %MMF and the other three studied parameters was insignificant, inverse, and weak. It should be remarked that T-Scan studies from the dental literature do not frequently address this parameter of digital occlusal analysis.

A new parameter followed in our study was *MAT-OTB*, which refers to the time elapsed from the last occlusal contact in static intercuspation (B line) to MI, characterized by the maximum area of tooth contact (i.e., the MA frame). The mandible slides on guiding cusps inclines from static intercuspation to MI. The value of the MAT-OTB depends on the morphology of the dental crowns and the location of the occlusal contacts in static intercuspation.

A shorter MAT-OTB means less time during which the inclines of the guiding cusps are bringing the mandible back into MI, under the heavy load created by the contraction of the elevator muscles. Because of the high occlusal forces, which are distributed across unstable occlusal contacts occurring between cusp slopes, we consider that the MAT-OTB should be as short as possible.

Because it is a new introduced parameter in our digital occlusal analysis, we did not have a normal value for MAT-OTB to interpret the results measured in NHP. In addition, the correlations between MAT-OTB and the other three studied parameters were not conclusive. Further studies on larger populations of healthy subjects with normal occlusal relationships could gather more data on this parameter. The normal values of MAT-OTB should be determined in conditions similar to those used by Kerstein to establish the normal values of OT and AOF.

## 5. Conclusions

The study investigated the influence of TNP on static occlusion in young healthy subjects with normal occlusal relationships. We used the T-Scan III system to record MI in standing subjects in NHP and TNP. While NHP was the logical starting point of investigations from a clinical point of view, it proved to be a major limitation of the T-Scan study. We added 30° to the neck flexion angle measured in NHP in order to simulate a TNP that was as close as possible to the reality, although this extreme flexion could induce issues during T-Scan registrations in TNP in standing patients. Thus, the OT, AOF, % MMF, and MAT-OTB values measured in TNP were not conclusive regarding the influence of this head and neck position on static occlusion. In conclusion, the NHP should not be used as the starting position of TNP simulations in T-Scan studies for standing subjects. Future research could avoid the above limitation by starting in a head posture with a neck flexion angle of 0°, measured with a Cervical Range of Motion device. Using a manual goniometer to measure the neck flexion angle was another limitation of our study.

However, the T-Scan analyses of static occlusion for standing healthy subjects with normal occlusal relationships from a clinical point of view revealed significant results in NHP. The high values of the AOF and OT in NHP described an occlusal instability that must be continuously compensated by the masticatory system of most subjects. The occlusal instability was further amplified by the fact that the correlation between OT and AOF was significant, direct, and strong in NHP. While our subjects do not have an increased risk for TMD based on the medium value of the AOF, they do have a higher risk to develop a less severe TMD based on the medium value of the OT in our study, as well as on the values advanced by Baldini for dysfunctional patients [37].

In conclusion, healthy standing subjects with normal occlusal relationships from the clinical point of view revealed an occlusal instability in NHP when examined with the T-Scan. Their masticatory system must continuously adapt to compensate for that occlusal instability, in order to avoid the “tipping” of the system towards dysfunction. The use of smartphones for different tasks (e.g., texting, internet surfing, etc.) increases the odds of awake bruxism, which consists in the forceful, involuntary, and unconscious clenching of teeth in MI, as stated in Section 4 [34,35]. This parafunction can overload the masticatory system, reducing its capacity to adapt to occlusal instability [20]. Therefore, even if healthy standing subjects are adopting a NHP while using smartphones, they should be advised not to touch their teeth except when swallowing.

The MAT-OTB, which refers to the time elapsed from the last occlusal contact in static intercuspation to MI, needs further research in order to gain more data on this new parameter.

## Figures and Tables

**Figure 1 medicina-58-01303-f001:**
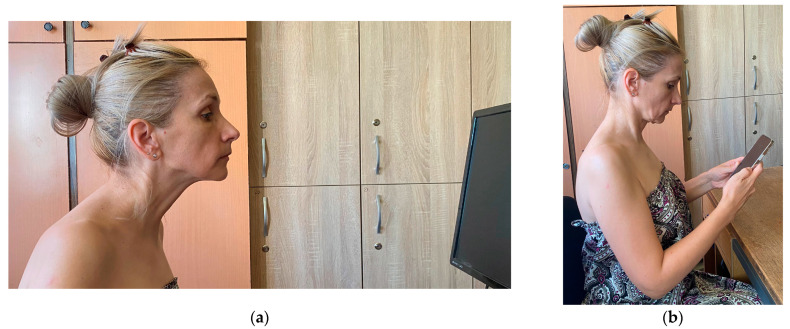
(**a**) Forward head posture (FHP), which is sometimes called the desktop neck posture; (**b**) text neck posture (TNP)—examples given by one of the examiners in the study.

**Figure 2 medicina-58-01303-f002:**
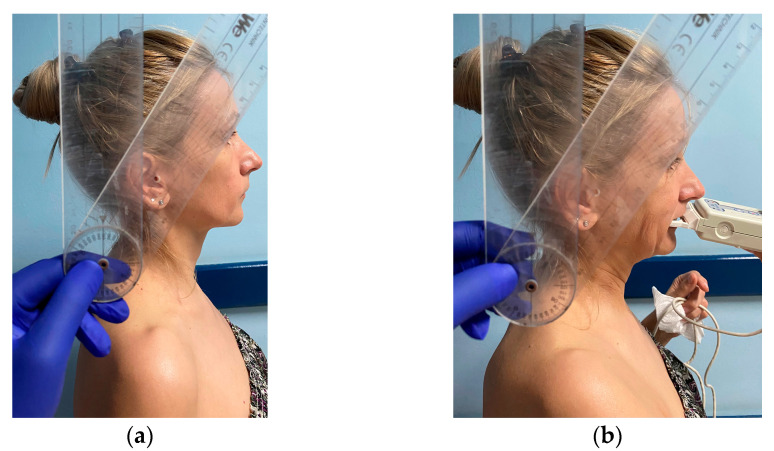
(**a**) Measurement of the neck flexion angle with a manual goniometer in NHP; (**b**) T-Scan registration in NHP.

**Figure 3 medicina-58-01303-f003:**
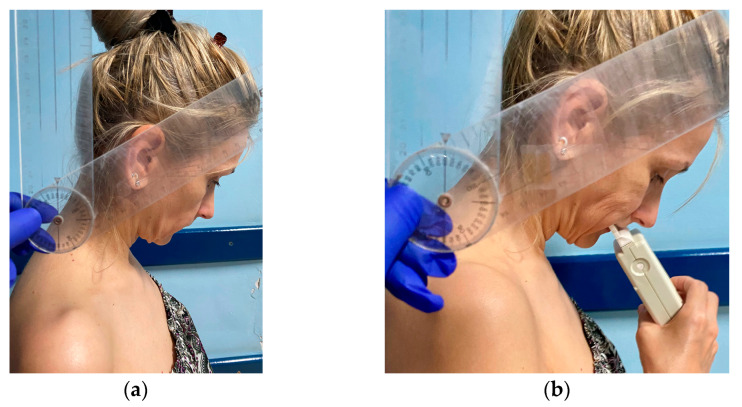
(**a**) Measurement of the neck flexion angle with a manual goniometer in TNP; (**b**) T-Scan registration in TNP.

**Figure 4 medicina-58-01303-f004:**
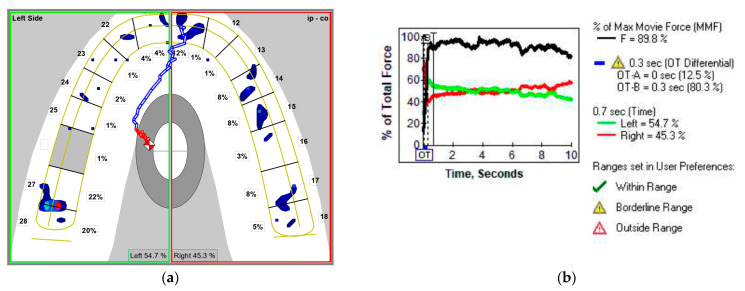
MA frame of the T-Scan registration of MC subject in NHP: (**a**) two-dimensional (2D) view with the distribution of the occlusal force between the left and the right side of the arch; (**b**) graph of the % of the total force in time, with %MMF, OT, OTB, and MAT (0.7 s).

**Figure 5 medicina-58-01303-f005:**
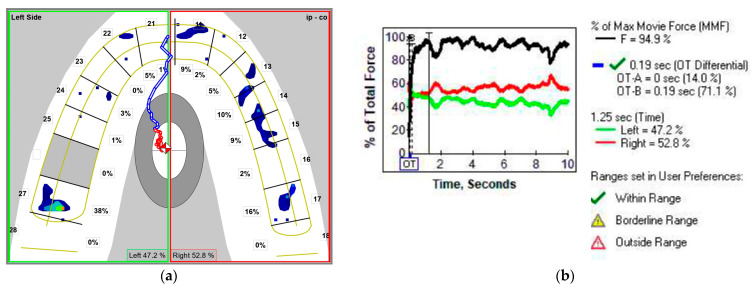
MA frame of the T-Scan registration of MC subject in TNP: (**a**) 2D view with the distribution of the occlusal force between the left and the right side of the arch; (**b**) graph of the % of the total force in time, with %MMF, OT, OTB, and MAT (1.25 s).

**Figure 6 medicina-58-01303-f006:**
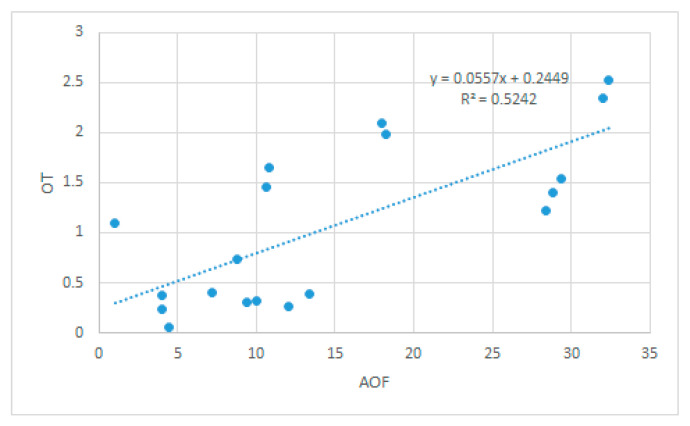
The graph that illustrates the direct correlation between OT and AOF in NHP.

**Table 1 medicina-58-01303-t001:** Demographic characteristics of the study group.

Variable	Characteristics	
**Gender—*n* (%)**	Male	4 (21.1%)
	Female	15 (78.9%)
**Age**		
Mean (standard deviation)	21.32 (0.95)	
Median (interquartile range)	21.00 (1.00)	

**Table 2 medicina-58-01303-t002:** Descriptive statistics and *p* values obtained from the applied Wilcoxon Signed Ranks Test.

Variable	Mean ± Standard Deviation	Median (Interquartile Range)	*p* ^1^
OTN	1.34 ± 1.84	0.41 (1.22)	0.658
OT30	1.32 ± 1.8	0.45 (1.39)	
AOFN	14.88 ± 10.39	10.8 (21.2)	0.344
AOF30	18.04 ± 12.87	17.1 (16.4)	
MMFN	97.5 ± 2.83	98.3 (2.3)	0.251
MMF30	96.31 ± 3.17	97.7 (4.9)	
MAT-OTBN	2.08 ± 1.82	1.95 (2.76)	0.117
MAT-OTB30	1.45 ± 2.3	0.85 (1.28)	

^1^ Statistically significant *p* < 0.05.

## Data Availability

The data supporting the reported results presented in this study are available within the article and in Appendix A.

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
