# Peer review of "Influence of the Text Neck Posture on the Static Dental Occlusion"

_medicina, 2022, doi:10.3390/medicina58091303_

Round 1

Reviewer 1 Report

Dear Authors,

The article is very well written and the study has scientific relevance. I congratulate the team for the high-quality work. I have one doubt: - Was a sample calculation performed or was the sample chosen for convenience? I suggest adding this information to the text.

Reviewer 2 Report

The paper addresses a potential issue with the extended use of personal phones .  The  method to assess he potential effect are to use a bite scan and compare several positions.

The research methods need to be improved as there is no calibration, no blind testing  and correlations between the results reported could be clarified. 

There is also no mention of the mandibular position when people are using their phone. It is possible that the teeth stay out of occlusion during phone use . This should have been tested prior to the study. 

Reviewer 3 Report

The theme of the research is very interesting and importante, since nowadays the head position while looking at the cell phone is deviated for a longer time during the day.

Introduction is well written.

Methods- I sholud aks about parafunction of the subjects evaluated. If they had any, that could be influencing the resulto f the study.

Also, was the cervical position evaluated? Although there was no pain or limited range of motion in the cervical spine, a patient with altered cervical position could be influencing the end result.

Was there any exclusion criteria? A patient with fibromialgia or autoimune disease, trauma or habit that could impact the cervical position.

Discussion did show the limitations of the study and discussed other studies in the same matter.

Round 2

Reviewer 2 Report

The results are not clearly stated. There are a lot of numbers but it would be nice to better explain their correlations and avoid using data that do not really add to the study.
